# A Model to Estimate the Effect of International Traffic on Malaria Cases: The Case of Japan from 1999 to 2021

**DOI:** 10.3390/ijerph19020880

**Published:** 2022-01-13

**Authors:** Hiroyuki Noda

**Affiliations:** 1Public Health, Department of Social Medicine, Graduate School of Medicine, Osaka University, Osaka 565-0871, Japan; hiroyuki-noda@umin.net; 2Cabinet Secretariat, Tokyo 100-8968, Japan

**Keywords:** international traffic, malaria, COVID-19, Japan

## Abstract

Aiming to identify the potentially reduced malaria cases by stagnation of international traffic after the COVID-19 pandemic, a longitudinal analysis of malaria cases as well as entries of Japanese and foreigners was conducted using data from 5 April 1999 to 30 September 2021 in Japan. Multivariable risk ratios were calculated with the Poison regression model as a predictive model of malaria cases by the number of entries for Japanese and foreigners. A generalized regression model was used to examine an association of time trend with entries for Japanese and foreigners using data before 2019, to estimate the potentially reduced number of entries after 2020. The potentially reduced number of malaria cases was estimated by the potentially reduced number of entries for Japanese and foreigners after 2020 using a multivariable Poison regression model. The multivariable risk ratio (95% confidence intervals) of malaria case numbers per 100,000 persons increment of entries per day was 3.41 (1.50–7.77) for Japanese and 1.47 (0.92–2.35) for foreigners. During 2020, a potential reduction of 28 (95% confidence limit: 22–34) malaria cases was estimated, which accounted for 58% (52–63%) of malaria cases in Japan. These finding suggest that the stagnation of international traffic during the COVID-19 pandemic reduced the number of malaria cases in Japan. This model may be helpful for countries without indigenous malaria to predict future trends of imported malaria cases.

## 1. Introduction

International traffic, which is defined as “the movement of persons, baggage, cargo, containers, conveyances, goods or postal parcels across an international border, including international trade” in International Health Regulation [1], has been at the heart of the debate on global health regulations and the weighting of quarantine measures and sanitary measures with the aim of prevention and control of imported infectious diseases for several centuries [2,3]. In line with historical frameworks, International Health Regulation states “The purpose and scope of these regulations are to prevent, protect against, control and provide a public health response to the international spread of disease in ways that are commensurate with and restricted to public health risks, and which avoid unnecessary interference with international traffic and trade” [1,4]; based on this spirit, interference with international traffic has rarely been introduced in the context of global health. However, the situation changed completely with the onset of Coronavirus disease 2019 (COVID-19) [5,6].

Since the lockdown of Wuhan following the spread of COVID-19 in the city, almost all countries have broadly applied interference with international traffic, such as international travel restrictions and border control [4,7]. An annual report by the World Tourism Organization (UNWTO) indicates that international tourist arrivals fell globally by 74%, from 1.46 billion arrivals in 2019 to 381 million arrivals in 2020 [8]. In Japan, the number of entries reported by the Immigration Services Agency of Japan reduced from 51 million persons in 2019 to 8 million persons in 2020 [9]. However, international travel is slowly picking up from very low levels in May 2021, after a long stagnation, as some destinations have started to ease travel restrictions and consumer confidence has risen slightly [10]. UNWTO indicated that international tourism saw a minor uptick in May 2021, with arrivals declining by 82% (versus May 2019), after falling by 86% in April [10]. This minor uptick may represent a path back to the normalization of international traffic, even though levels remain far from those in the pre-pandemic era.

International traffic is seen as one of the main human influences on the international spread of infectious diseases in the twenty-first century [4]. For example, indigenous malaria was eradicated in 1961 [11,12,13], and malaria is currently one of the imported infectious diseases in Japan. Malaria cases were reported annually in the range around 60 cases between 2006 and 2017 in Japan, and almost all cases were reported as imported malaria [14]. Nevertheless, after the beginning of the COVID-19 pandemic, the number of malaria cases has reduced significantly as a result of the stagnation of international traffic due to countermeasures against COVID-19. In Japan, the cumulative number of malaria cases was only 20 cases by the end of the epidemiological weeks in 2020 [15]. However, as this change may be caused by an association between international traffic (i.e., the number of entries crossing international borders) and imported infectious disease, imported infectious diseases such as malaria may increase again in the future due to the normalization of international traffic.

In Japan, malaria was eradicated, and almost all cases are reported as imported malaria [14]. Persons aged 20–39 years account for 62% of cases [14], and the report by the National Institute of Infectious Diseases concluded that this age distribution of malaria cases was associated with the countries representing malaria risk and the age distribution of oversea travelers [14]. This report also showed that 688 cases of the 704 cases between 2006 and 2017 were imported malaria, and persons who travelled to African countries accounted for 65% of cases [14]. These data suggest that malaria is rare in Japan, and a previous study suggested that there are few medical doctors with experience in diagnosing malaria in Japan [16]. Moreover, residents in countries without indigenous malaria, including Japan, are usually non-immune, as acquisition of immunity is slow and requires repeated exposure, and people with non-immunity are more likely to become severe cases [17]. Therefore, it may be important to increase prior probability in diagnosing malaria for preventing delay of diagnosis, and prediction of future trends of malaria cases may assist medical doctors in their awareness of malaria risks in countries without indigenous malaria, including Japan.

Globally, there were an estimated 241 million malaria cases in 2020 in 85 malaria-endemic countries, increasing from 227 million in 2019, with most of this increase coming from countries in the World Health Organization (WHO) Africa Region [18]. The latest report by the WHO indicates that 29 countries, mainly in Africa, accounted for 96% of malaria cases globally, and that six countries (i.e., Nigeria, the Democratic Republic of the Congo, Uganda, Mozambique, Angola, and Burkina Faso) accounted for about 55% of all cases [18]. On 6 October 2021, the WHO announced that it will be recommending widespread use of the RTS,S/AS01 (RTS,S) malaria vaccine, which is the first parasite vaccine to have obtained regulatory approval, for children in sub-Saharan Africa and in other regions with moderate-to-high *Pla**smodium Falciparum* transmission [19]. This innovative technology may reduce child mortality in high-risk malaria areas in the future, but these statistics show that African countries currently have most of the malaria cases in the world.

During the COVID-19 pandemic, the government of Japan has implemented risk-based border control by quarantine measures (e.g., submission of a certificate of a negative test result conducted within 72 h prior to departing the country/region visited, staying for specified days at facilities designated by the Chief of the Quarantine Station, and staying for the remaining period of 14 days after entry into Japan at places such as one’s own residence), denial of permission to enter for foreign nationals from specified countries/areas, and suspension of visa validity. On 31 January 2020, Hubei Province was designated as the first region for denial of permission to enter for foreign nationals, after declaration of a Public Health Emergency of International Concern (PHEIC) under the International Health Regulations [1]; this was subsequently expanded to 73 countries/areas on 3 April 2020, after the announcement of a pandemic by the WHO Director-General. At the end of 2021, new entry of foreign nationals from all countries/areas was suspended for the time being, as an emergency precautionary measure from a preventive perspective against the Omicron variant. These border control measures affect the number of people crossing the international borders of Japan and thus the imported malaria cases during the COVID-19 pandemic.

Thus, aiming to identify the potentially reduced malaria cases due to the stagnation of international traffic during the COVID-19 pandemic, a longitudinal analysis was conducted using the association of entries for Japanese and foreigners with malaria cases during the past two decades in Japan.

## 2. Materials and Methods

### 2.1. Study Materials

A longitudinal analysis of malaria cases as well as entries for Japanese and foreigners was conducted using data between 5 April 1999 and 30 September 2021 in Japan.

Monthly data of entries for Japanese and foreigners were obtained from Statistics on Legal Migrants. The statistics on international migration are compiled monthly and annually by the Immigration Services Agency of Japan based on reports submitted by Regional Immigration Services Bureaus and their branches and report the number of all entries crossing international borders of Japan (i.e., airports and seaports). Japanese was defined as “a person with Japanese nationality”, and foreigner was defined as “a person without Japanese nationality” who was not “a person under the agreement” in this article. “A person under the agreement” is defined as “the foreign military forces, military civilian and their families who entered into or departed from Japan by civilian ships or aircrafts under the agreements relating to the status of U.S. and U.N. military forces stationed in Japan” in Statistics on Legal Migrants.

Weekly data of malaria cases were obtained from reports by the National Institute of Infectious Diseases and the Ministry of Health, Labour and Welfare (MHLW). In Japan, medical doctors who diagnosed malaria must “file a notification with the prefectural governor via the chief of the nearest public health center immediately”, and the statistics of the infectious diseases are compiled weekly and annually by the National Institute of Infectious Diseases and the MHLW under the legal framework of the Act on the Prevention of Infectious Diseases and Medical Care for Patients with Infectious Diseases, namely the Infectious Diseases Control Act as of April 1999 [20]. Under the Infectious Diseases Control Act, malaria is defined as “disease caused by single infection or mixed infection of protozoans of the *Plasmodium* species such as *Plasmodium vivax*, *Plasmodium falciparum*, *Plasmodium malariae* and *Plasmodium ovale*” [20].

The population in Japan was obtained from Population estimates by the Statistics Bureau of Japan for the population as of the first day of each month and linearly imputed for daily data.

In this study, publicly available data were used from websites and published reports. As these publicly available data were anonymized, informed consent was not required.

### 2.2. Statistical Analysis

Crude and multivariable risk ratios (RRs) and 95% confidence intervals (95%CIs) were calculated with a Poison regression model adjusted for time trend (year) and seasonality (month), as a predictive model of malaria cases per capita by the number of entries for Japanese and foreigners, using daily data from 5 April 1999 to 30 September 2021. Daily data were calculated using weekly data of malaria cases and monthly data of entries as uniform in each range. As the global reduction of malaria cases [21] may affect the number of imported malaria cases in Japan, time trend (year) was included as a potential confounding factor in the multivariable Poison regression model. Similarly, seasonality (month) was also included as a potential confounding factor.

These RRs and 95%CIs were calculated in the two models. The first model included the number of entries for Japanese and for foreigners separately (Model 1), and the second model included both (Model 2). Whereas the presence of multicollinearity does not affect the efficiency of extrapolating the fitted model [22], the variance inflation factors (VIF) of these predictive variables were calculated to diagnose the multicollinearity [23]. VIFs above 5 were considered as the presence of multicollinearity [23]. This point of cutoff for VIFs is usually used for the diagnosis of multicollinearity, and VIFs above 5 mean that the R^2^ is 0.8 and above. When the R^2^ is 1, it means perfect collinearity [22,23].

Because both entries and malaria cases were largely reduced after 2020, this large reduction may affect the association of entries with malaria cases between 1999 and 2021. Therefore, RRs (95%CI) were also calculated after the exclusion of data after 2020 for sensitivity analysis.

As of the end of 2021, the number of malaria cases by gender prior to 2019 was publicly available. Therefore, a gender-specific analysis was also conducted using data prior to 2019.

A generalized regression model was used to examine the association of time trend (year) with the number of entries for Japanese and foreigners, adjusted for seasonality (month), using data up until 31 December 2019, to estimate the potentially reduced number of entries from 1 January 2020 to 30 September 2021. Because the number of entries of foreigners was increased from around 2012 by a national campaign based on the Tourism-based Country Promotion Act [24], a generalized regression model with multiple interaction terms of time trend [25] was used to examine the association with entries of foreigners. Statistics on Legal Migrants shows that the increased number of entries from around 2012 was caused by an increase of foreign tourists [9].

The potentially reduced number of malaria cases was estimated by the potentially reduced number of entries for Japanese and foreigners after 2020, using a multivariable Poison regression model. The 95% confidence limits for the potentially reduced number were calculated using the bootstrap method [26].

All statistical tests were 2-sided and conducted using SAS, version 9.4 (SAS Institute, Cary, NC, USA). P values below 0.05 were considered as statistically significant.

## 3. Results

A total of 1534 malaria cases were identified between 5 April 1999 and 30 September 2021. The weekly number of malaria cases was nine in the peak of 2000, but the respective number was four in the peak of 2019. In data before 2019, the percentage of male cases was 76% (1143 male cases and 353 female cases). The number of entries was largely reduced from 2019 to 2020 (Figure 1). The mean value (range) in the number of entries was 1.67 (1.27–2.06) million persons per month in 2019 and 0.31 (0.015–1.61) million persons per month in 2020 (*p* < 0.0001) for Japanese, and the respective value was 2.60 (2.21–2.89) million persons per month and 0.36 (0.004–2.70) million persons per month (*p* < 0.0001) for foreigners.

Table 1 shows that the high number of Japanese entries was associated with an increased number of malaria cases. The crude RR of malaria case numbers per 100,000 persons increment of entries per day was 7.95 (4.84–13.06), *p* < 0.0001, for Japanese and 0.60 (0.48–0.75), *p* < 0.0001, for foreigners in Model 1, and the respective multivariable RR (95%CI) was 5.24 (2.77–9.92), *p* < 0.0001, and 2.34 (1.65–3.30), *p* < 0.0001. In Model 2, the respective multivariable RR (95%CI) was 3.41 (1.50–7.77), *p* = 0.003, and 1.47 (0.92–2.35), *p* = 0.10, which indicates that there was no association of the number of entries for foreigners after adjustment in Model 2, whereas there was no variable with ≥5 VIF. The RRs were not altered substantially after exclusion of the data after 2020.

These associations were not substantially altered in gender-specific analysis using data before 2019. The multivariable RR of malaria case numbers per 100,000 persons increment of entries per day was 21.87 (6.90–69.27), *p* < 0.0001, for Japanese and 2.05 (1.19–3.54), *p* = 0.01, for foreigners in Model 2 for male cases, and the respective multivariable RR (95%CI) was 18.73 (2.46–142.32), *p* < 0.005, and 1.38 (0.50–3.87), *p* = 0.54, for female cases.

Figure 2 shows the potentially reduced number of malaria cases due to stagnation of international traffic after 2020. The total of 28 (95% confidence limits: 22–34) malaria cases per year was estimated to be potentially reduced during 2020 in Japan, which accounted for 58% (52–63%) of malaria cases. The respective number was 24 (17–31) malaria cases per year during 2021, which accounted for 50% (41–56%).

## 4. Discussion

The association of the number of entries for Japanese and foreigners with the number of malaria cases during the past two decades in Japan was examined, aiming to identify the potentially reduced number of malaria cases due to stagnation of international traffic after the COVID-19 pandemic. The number of entries for Japanese was strongly associated with the number of malaria cases, and 58% of malaria cases were estimated to be potentially reduced during 2020 in Japan. These findings suggest that the stagnation of international traffic during the COVID-19 pandemic reduced the number of malaria cases in Japan.

This article indicated that the stagnation of international traffic during the COVID-19 pandemic potentially reduced imported malaria cases. This finding indicates the possibility of a future rebound of malaria cases after the normalization of international traffic. In Japan, almost all medical doctors have little experiences with malaria diagnosis [16] and may have difficulty in early detection due to a lack of diagnostic experience in a clinical setting and reduced prior probability [5]. A previous study suggested that less information about malaria risks in overseas travelers might lead to a delay in diagnosing malaria, as there are few medical doctors with experience in diagnosing malaria in Japan [16]. Before normalization of international traffic, we need to raise the awareness of medical doctors, which will help them to diagnose patients by increasing prior probability. 

This article showed that the number of entries for Japanese was strongly associated with the number of malaria cases. In Japan, malaria was eradicated [13], and almost all cases are recognized as an imported infectious disease [14]. If infected overseas, most Japanese residents may be delayed in going to the hospital as they would not suspect malaria infection [16]. Moreover, overseas travelers infected with malaria may be more likely to become severe cases, because protection from malaria is acquired from repeated exposure [17]. As naturally acquired immunity to malaria is usually developed by individuals living in a malaria-endemic areas [17], visitors infected with malaria while overseas are more likely to be non-immune. A previous study indicates that the risk of severe cases is predominantly confined to malaria-naive visitors as well as small children and pregnant women in areas of high transmission [17]. Moreover, a previous study suggested that health education before overseas travel may reduce “patient’s delay” and “doctor’s delay” in Japan, which may help patients to access early diagnosis and treatment [16]. Earlier access to diagnosis leads to better outcomes in a clinical setting. A previous study suggested that most delays in fatal cases were in seeking care [27]. We also need to raise awareness and to promote health education on malaria risks as well as chemoprophylaxis for overseas travelers after normalization of international traffic. 

On the other hand, there was no association of the number of entries for foreigners with the number of malaria cases after adjustment of time trends. This reduced risk was due to the association with the time trend, whereas there was not multicollinearity. In fact, the number of entries for foreigners increased slightly between 1999 and 2011, and then the number began to increase rapidly after 2012. Despite this change, the time trend of malaria cases was not changed around 2012. This indicates that there may be almost no impact of foreigners’ entry itself on the number of malaria cases in Japan.

One of the plausible reasons for why the entry for foreigners was not associated with malaria cases may be the distribution of nationality in the number of entries in Japan. Statistics on Legal Migrants in 2019 showed that the percentage of entries was high for foreigners from East Asian countries such as China and South Korea [9], and the population in these countries has a low risk of malaria infection [18]. In fact, China was classified as malaria free as of 2000 [18], and there were no indigenous cases after 2017. South Korea is classified as a country with indigenous cases, but the number of indigenous malaria cases was only 485 in 2019 [18]. On the other hand, 29 countries, mainly in Africa, account for 96% of malaria cases globally, and six countries account for about 55% of all cases [18]. The percentage of entries for foreigners from the 29 countries was only 0.7% during 2019 in Japan, and the respective percentage from the six countries was 0.02%.

In Japan, malaria was eradicated, and the number of cases reported annually before the COVID-19 pandemic was around 60 [11,12,13,14], which suggests that malaria is rare in Japan. However, residents of Japan are usually non-immune, and persons with non-immunity are more likely to become severe cases [17]. Moreover, a previous article suggested that there was “patient’s delay” and “doctor’s delay” in Japan, as there are few medical doctors with experience in diagnosing malaria [16]. Therefore, it may be important to increase prior probability in diagnosing malaria for preventing delay of diagnosis, and the future trend of malaria cases predicted in this article can greatly assist medical doctors to be aware of malaria risks. 

This article showed a predictive model to estimate the effect of international traffic on malaria cases. Although the association was estimated using data in Japan, it may be possible to conduct similar analysis in other countries without indigenous malaria, because these data (i.e., the number of entries crossing international borders and malaria cases) are basic statistics that are usually available in each country. The predictive model using data in each country may assist medical doctors to be aware of malaria risk, and to promote health education as well as chemoprophylaxis for overseas travelers from countries without indigenous malaria.

Public health measures always present side effects beyond the main effect on the target disease [5], because they are strongly related to society. For example, it has been pointed out that the countermeasures against COVID-19 have affected social connectedness as well as employment, which may lead to other health issues such as dementia and suicide [28,29,30]. Furthermore, a previous study showed that the countermeasures against COVID-19, such as domestic travel restrictions and school closures, may reduce hospitalization by acute respiratory infection and respiratory syncytial virus [31]. This article showed that the stagnation of international traffic as a side effect of the countermeasures against COVID-19 reduced the number of malaria cases in Japan.

This article showed that the stagnation of international traffic reduced malaria cases during 2020 in Japan, but the latest world malaria report indicates that the global incidence rate of malaria has risen slightly from 56 cases per 1000 population at risk in 2019 to 59 cases per 1000 population at risk in 2020 [18]. This report also indicates that the decline of the incidence rate has stalled in several countries with moderate or high transmission since 2015, and that the situation was made worse, especially in sub-Saharan Africa, by the COVID-19 pandemic and other humanitarian emergencies [18]. As a reason, this report suggests that the supply chain systems for health commodities have experienced disruptions in transportation from the site of manufacturing to countries and within countries. Public health policy as well as global infectious disease and the economy are strongly related to society, and the effect sometimes becomes pleiotropic. The stagnation of international traffic by countermeasures against COVID-19 may have reduced imported malaria cases in Japan through the decline of Japanese entries, but it may also have increased indigenous malaria cases, especially in sub-Saharan Africa, through disruptions of the supply chain systems for health commodities. These findings indicate the importance of paying attention to pleiotropic effects when implementing public health policy.

A limitation of this article is that individual data of malaria cases, such as age and the countries to which infected persons travelled, were not used, because these data are not publicly available. A summary of malaria cases between 2006 and 2017 reported by the National Institute of Infectious Diseases showed that people aged 20–39 years accounted for 62% of cases, and it concluded that this age distribution of malaria cases was associated with the countries to which infected persons travelled and the age distribution of overseas travelers [14]. This report also showed that 688 of 704 cases were imported malaria, and that persons who travelled to African countries accounted for 65% of cases [14]. Second, as there are not statistics on the purpose of international travel (e.g., leisure or business) for Japanese travelers in Statistics on Legal Migrants, the association with the purpose of international travel was not examined in this study. Because a previous study showed that malaria cases who had traveled to countries with endemic malaria for tourism had higher mortality rates compared to those for visiting friends and relatives [27], the purpose of international travel may also affect the risk of infection. As the individual data of entries as well as malaria cases may be helpful to make the predictive model more accurate, further studies using individual data are required for more accurate predictive models. Third, as the data of malaria cases specified by gender after 2020 were not publicly available in Japan as of the end of 2021, the gender-specific association of entries with malaria cases was not examined using data after 2020. However, the gender-specific association was not substantially altered when using data prior to 2019, which suggests that there may be no gender differences.

For strength, to the author’s knowledge, this is the first report on the effect of international traffic during the COVID-19 pandemic on imported infectious diseases. This article can be helpful to raise awareness regarding the future increase of imported infectious diseases as rebounded by the normalization of international traffic.

## 5. Conclusions

This article provided epidemiological evidence that the stagnation of international traffic due to countermeasures against COVID-19 potentially reduced malaria cases by 58% in 2020 in Japan. International traffic was stagnated due to the COVID-19 pandemic, but international travel is slowly picking up from very low levels. We need to promote the preparation for the rebound of imported malaria cases in response to the normalization of international traffic.

## Figures and Tables

**Figure 1 ijerph-19-00880-f001:**
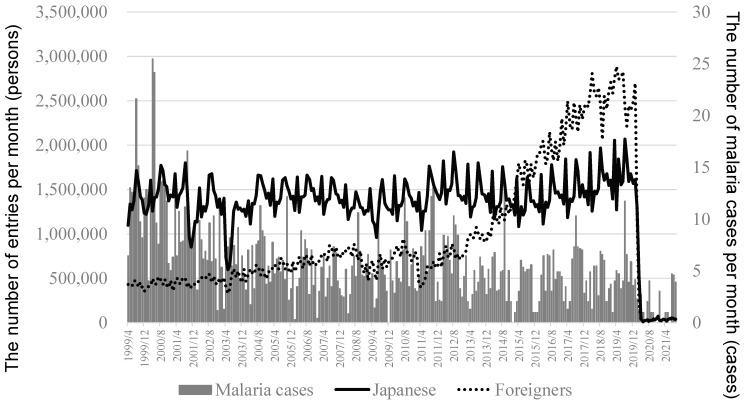
Secular trend in the number of malaria cases and entries for Japanese and foreigners in Japan. The mean value (95% confidence intervals) in the number of entries per month was 1.32 (1.31–1.33) million persons for Japanese between April 1999 and September 2021, and the respective value was 0.97 (0.95–0.98) million persons for foreigners. The bend point for the number of entries for foreigners was on 24 October 2012.

**Figure 2 ijerph-19-00880-f002:**
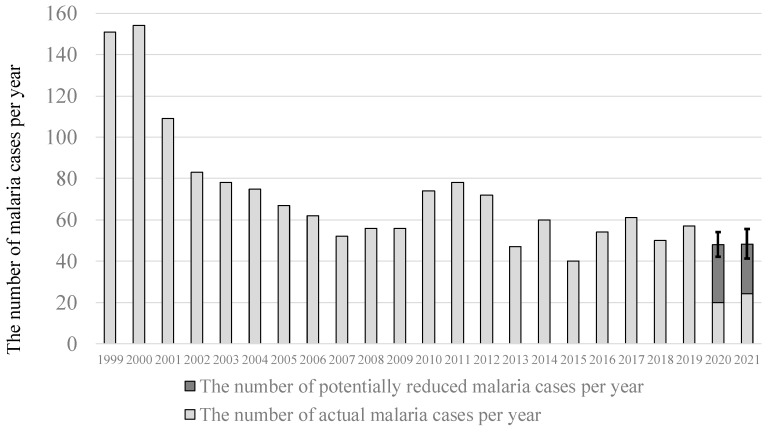
Secular trend in the actual number and the potentially reduced number of malaria cases in Japan. The error bars show 95% confidence limits of the number of potentially reduced malaria cases per yar.

**Table 1 ijerph-19-00880-t001:** Risk ratio (RR) of malaria case numbers per 100,000 persons increment of entries for Japanese and foreigners per day.

	Risk Ratio Per 100,000 Persons Increment Per Day			
	By 30 September 2021		By 31 December 2019	
	Model 1 ^a^	Model 2 ^b^	Model 1 ^a^	Model 2 ^b^
Entries for Japanese				
Crude RR	7.95 (4.84–13.06) ^d^	20.73 (12.00–35.82) ^d^	3.49 (1.65–7.38) ^d^	10.44 (4.70–23.17) ^d^
Multivariable RR ^c^	5.24 (2.77–9.92) ^d^	3.41 (1.50–7.77) ^d^	7.17 (2.32–22.13) ^d^	5.46 (1.72–17.32) ^d^
Entries for Foreigners				
Crude RR	0.60 (0.48–0.75) ^d^	0.31 (0.24–0.40) ^d^	0.39 (0.30–0.50) ^d^	0.31 (0.24–0.41) ^d^
Multivariable RR ^c^	2.34 (1.65–3.30) ^d^	1.47 (0.92–2.35)	1.89 (1.17–3.04) ^e^	1.57 (0.96–2.55)

^a^ Model 1 included either the number of entries for Japanese or that for foreigners. ^b^ Model 2 included the number of entries for both Japanese and foreigners. ^c^ Further adjusted for months and years. ^d^
*p* < 0.01. ^e^
*p* < 0.05.

## Data Availability

Publicly available datasets were analyzed in this study. Data on entries for Japanese and foreigners and population statistics can be found at https://www.e-stat.go.jp/ (access on: 5 December 2021). Data on malaria cases can be found at https://www.niid.go.jp/niid/ja/idwr.html (access on: 5 December 2021).

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
