# Peer review of "A Model to Estimate the Effect of International Traffic on Malaria Cases: The Case of Japan from 1999 to 2021"

_ijerph, 2022, doi:10.3390/ijerph19020880_

Round 1
Reviewer 1 Report
I think the authors have done a very elaborate study on a topic that is really not a problem in Japan. However, the analysis model could be used by other people living in areas where malaria is a health problem.

Author Response
I think the authors have done a very elaborate study on a topic that is really not a problem in Japan. However, the analysis model could be used by other people living in areas where malaria is a health problem.
>>Thank you very much for valuable comments. I revised my manuscript point by point as follows.
An alternative title could be: A model to estimate the effect of international stagnation in malaria cases, the case of Japan. (Title)
>> I revised the title as follows; “A model to estimate the effect of international traffic on malaria cases: a case of Japan from 1999 to 2021”
I think the author should include some comments on the impact of his predictive model beyond Japan. (Abstract)
>> I added the sentence “This model may be helpful for countries with indigenous malaria to predict future trend of im-ported malaria cases.” in abstract.
Apparently, Japan does not have a malaria problem. Therefore, the author must better justify their study. Please see the following comment. (L51-52)
>>I added the paragraph “In Japan, malaria was eradicated, and almost all cases were reported as imported malaria [14]. Persons aged 20-39 years accounted for 62% of cases [14], and the report by National Institute of Infectious Diseases concluded that this age distribution of malaria cases was associated with countries persons who infected malaria travelled to and age distribution of oversea travelers [14]. This report also showed that 688 cases of 704 cases between 2006 and 2017 were imported malaria, and persons who travelled to African countries accounted for 65% of cases [14]. These data suggest that malaria is rare in Japan, and a previous study suggested that there are few medical doctors with experience in diagnosing malaria in Japan [16]. Moreover, residents in countries without indigenous malaria including Japan are usually non-immune, as acquisition of immunity is slow and requires repeated exposure, and persons with non-immune are more likely to become severe cases [17]. Therefore, it may be important to increase prior probability in diagnosing malaria for preventing delay of diagnosis, and prediction in future trend of malaria cases may assist medical doctors to aware of malaria risks in countries without indigenous malaria including Japan.” in L67-L81, to justify my article.
I think the author should emphasize the global importance of his study, describing that it takes Japan only as an example. (Introduction)
>>I added the sentence “prediction in future trend of malaria cases may assist medical doctors to aware of malaria risks in countries without indigenous malaria including Japan.” In L79-L81.
The author has partially established the importance of his study beyond Japan. I only suggest to emphasize the importance of the proposed model for estimating malaria cases in different areas and conditions. (Discussion)
>>I added the paragraph “This article showed a predictive model to estimate the effect of international traffic on malaria cases. Although the association was estimated using data in Japan, it may be possible to conduct similar analysis in other countries without indigenous malaria, because these data (i.e. the number of entries crossing international border and malaria cases) are basic statistics which are usually available in each country. The predictive model using data in each country may assist medical doctors to aware of malaria risks, and to promote health education as well as chemoprophylaxis for overseas travelers in countries without indigenous malaria.” in L289-L296.
Reviewer 2 Report
Dear Author,
It is an interesting study that corresponds to the scope of the International Journal of Environmental Research and Public Health and may be suitable for Journal’s profile, however I have specified some comments on the article:
- In the Introduction section it would be helpful to give more background information about epidemiology of malaria worldwide with providing source references. It is advisable to extend statement and give more details in the lines 185-190, and then to remove them from the Discussion section. In addition, the introduction section should describe in more detail the current state of art regarding vaccinations against malaria.
- The Material and Methods section: malaria should be defined on the basis of ICD-10 (standard classification will be useful).
- In the Results: relevant information on the prevalence of malaria by age and sex categories of the studied groups would be necessary, and also description of the territorial division of the disease is needed. These points should be placed in the description of Material and Methods, as well as in the Discussion. Furthermore, Figures 1 and 2 need a more accurate and informative comments on trends.
- There should be more well-balanced discussion about the limitations and strengths of the study.
- There are some typographical/punctuation errors in the manuscript that should be corrected e.g., 58%(52%-63%) should be changed into 58% (52%-63%).
- Please, highlight the changes to the revised version of your manuscript using a different color / or a marked script.
- Important: Please note that current version of the manuscript is too short regarding Journal’s style and requirement, which is preordained minimum as 3000 words. Thus, a modification and a careful revision of the text length is necessary after all the above points have been addressed.
Author Response
Dear Author,
It is an interesting study that corresponds to the scope of the International Journal of Environmental Research and Public Health and may be suitable for Journal’s profile, however I have specified some comments on the article:
>>Thank you very much for valuable comments. I revised my manuscript point by point as follows.
In the Introduction section it would be helpful to give more background information about epidemiology of malaria worldwide with providing source references. It is advisable to extend statement and give more details in the lines 185-190, and then to remove them from the Discussion section.
>> I added the paragraph “Globally, there were an estimated 241 million malaria cases in 2020 in 85 malaria endemic countries, increasing from 227 million in 2019, with most of this increase coming from countries in the WHO Africa Region [18]. The latest report by WHO indicates that 29 countries mainly in Africa accounted for 96% of malaria cases globally, and that 6 countries (i.e. Nigeria, the Democratic Republic of the Congo, Uganda, Mozambique, Angola and Burkina Faso) accounted for about 55% of all cases [18]. On October 6th, 2021, WHO announced that it will be recommending widespread use of the RTS,S/AS01 (RTS,S) malaria vaccine, which is the first parasite vaccine to have obtained regulatory approval, for children in sub-Sahara Africa and in other regions with moderate-to-high Plasmodium Falciparum transmission [19]. This innovative technology may reduce child mortality in high-risk area of malaria in the future, but these statistics show that African countries currently have most of the malaria cases in the world.” in L82-L93.
In addition, the introduction section should describe in more detail the current state of art regarding vaccinations against malaria.
>>I added the some sentences “On October 6th, 2021, WHO announced that it will be recommending widespread use of the RTS,S/AS01 (RTS,S) malaria vaccine, which is the first parasite vaccine to have obtained regulatory approval, for children in sub-Sahara Africa and in other regions with moderate-to-high Plasmodium Falciparum transmission [19].” in L87-L91.
The Material and Methods section: malaria should be defined on the basis of ICD-10 (standard classification will be useful).
>>Because the definition of infectious diseases in the Infectious Disease Control Act in Japan does not use ICD code, I added the following sentence; “Under the Infectious Diseases Control Act, malaria is defined as “disease caused by single infection or mixed infection of protozoans of the Plasmodium species such as Plasmo-dium vivax, Plasmodium falciparum, Plasmodium malariae and Plasmodium ovale” [20].” in L136-L139.
In the Results: relevant information on the prevalence of malaria by age and sex categories of the studied groups would be necessary, and also description of the territorial division of the disease is needed. These points should be placed in the description of Material and Methods, as well as in the Discussion. Furthermore, Figures 1 and 2 need a more accurate and informative comments on trends.
>>Because individual data of malaria cases such as age and countries persons who infected malaria travelled are not publicly available, I could not use these data. Therefore, I added the limitation as follows: ” A limitation of this article is that individual data of malaria cases such as age and countries persons who infected malaria travelled to were not used, because these data are not publicly available. A summary of malaria cases between 2006 and 2017 reported by National Institute of Infectious Diseases showed that people aged 20-39 years ac-counted for 62% of cases, and concluded that this age distribution of malaria cases was associated with countries persons who infected malaria travelled to and age distribution of oversea travelers [14]. This report also showed that 688 cases of 704 cases were imported malaria, and that persons who travelled to African countries accounted for 65% of cases [14].” in L322-329. Moreover, because data of the number of gender-specific malaria cases after 2020 are not publicly available as of the end of 2021, I added the result as well as method and limitation of gender specific analysis using data before 2019 as follows; “As of the end of 2021, the number of malaria cases by gender before 2019 was publicly available. Therefore, the gender-specific analysis was also conducted using data before 2019.” in L169-L171 as method, “In data before 2019, the percentage of male cases was 76% (1,143 male cases and 353 female cases).” in L190-L191, “These associations were not substantially altered in gender-specific analysis using data before 2019. The multivariable RR of malaria case numbers per 100,000 persons increment of entries per day was 21.87 (6.90-69.27), p<0.0001 for Japanese and 2.05 (1.19-3.54), p=0.01 for foreigners in model 2 for male cases, and the respective multivariable RR (95%CI) was 18.73 (2.46-142.32), p<0.005 and 1.38 (0.50-3.87), p=0.54 for female cases.” in L212-L216 as the result and “Third, as data of malaria cases specified by gender after 2020 were not publicly available in Japan as of the end of 2021, the gender-specified association of entries with malaria cases was not examined using data after 2020. However, the gender-specified association was not substantially altered using data before 2019, which suggests that there may be no gender differences.” in L337-L341 as limitation.
There should be more well-balanced discussion about the limitations and strengths of the study.
>> I added the paragraphs of limitations and strength as follows; “A limitation of this article is that individual data of malaria cases such as age and countries persons who infected malaria travelled to were not used, because these data are not publicly available. A summary of malaria cases between 2006 and 2017 reported by National Institute of Infectious Diseases showed that people aged 20-39 years ac-counted for 62% of cases, and it concluded that this age distribution of malaria cases was associated with countries persons who infected malaria travelled to and age distribution of oversea travelers [14]. This report also showed that 688 cases of 704 cases were imported malaria, and that persons who travelled to African countries accounted for 65% of cases [14]. Second, as there is not statistics in purpose of international travel (e.g. leisure or business) for Japanese in Statistics on Legal Migrants, the association specified by purpose of international travel was not examined in this study. Because a previous study showed that malaria cases who traveled to countries with endemic malaria for tourism had higher mortality rate, compared to those for visiting friends and relatives [27], the purpose of international travel may also affect risk of infection. As individual data of entries as well as malaria cases may be helpful to make the predictive model more accurate, further studies using individual data are required for more accurate predictive models. Third, as data of malaria cases specified by gender after 2020 were not publicly available in Japan as of the end of 2021, the gen-der-specified association of entries with malaria cases was not examined using data after 2020. However, the gender-specified association was not substantially altered using data before 2019, which suggests that there may be no gender differences.
For strength, to my knowledge, this is the first report of the effect of international traffic in the COVID-19 pandemics on imported infectious diseases. I believe that this article would be helpful to aware the future increment of imported infectious diseases as rebound by normalization of international traffic.” in L322-L345.
There are some typographical/punctuation errors in the manuscript that should be corrected e.g., 58%(52%-63%) should be changed into 58% (52%-63%).
>> I revised these errors and such.
Please, highlight the changes to the revised version of your manuscript using a different color / or a marked script.
>> I highlighted the changes using color.
Important: Please note that current version of the manuscript is too short regarding Journal’s style and requirement, which is preordained minimum as 3000 words. Thus, a modification and a careful revision of the text length is necessary after all the above points have been addressed.
>> I revised my manuscript, and the length of revised manuscript is currently 4,828 words.
Reviewer 3 Report
While this article reveals some interesting finding around the importation of malaria into Japan, the article requires significant revision before it can be considered for publication. I have listed a few comments below to assist with the revision. I also recommend that the article be reviewed by a English scientific editor before re-submission to ensure rules of scientific writing and English grammar have been adhered to.
- Pg1 Ln1-2: The article title needs to be reworded to more clearly articulate the aim of the article. I do not understand the term “international travel stagnation” and think most readers will not understand the phrase either. A time frame should be added to the title as well.
- Define what you mean by “foreigners”. Are they tourists, students or non-Japanese nationals working in Japan?
- The introduction needs to be revised to ensure the readers have a clear idea of the malaria situation in Japan, whether malaria importation is a problem, the routes of malaria importation, COVID travel restrictions in Japan, what the research described in this article aimed to achieve and these possible uses of research findings. Terms like international traffic need to be explained – does this refer to cargo, business or leisure travel?
- Pg1 Ln28: I do not understand what “the genealogy of these frameworks” means. Please rephrase more clearly.
- Pg1 Ln35: We are still very much in the midst of a COVID-19 pandemic – so please correct your statement about the being in the post-Corona era.
- Pg1 Ln36: I am assuming the word “all” is missing before the word “countries”
- Pg1 Ln39-40: Please stated whether these travel figures are for Japan, Asia or the global.
- Pg1 Ln43-45: These relaxation in international travel was it leisure or business? What do you mean by minor uptick?
- Pg2 Ln52: Please clarify what you mean by “ the identified area where cases infected have been estimated as oversea.”
- Pg2 Ln58: What proof do you have that malaria cases will increase in Japan once covid-related travel restrictions are lifted?
- Pg2 Ln65: Where these entries for all international borders: land, air and sea?
- Pg2 Ln81: Please add that as these publicaly available data were anonymized, informed consent was not required.
- Pg3 Ln98: What guided the decision that VIFs above 5 indicted multicollinearity?
- Pg3 Ln100: Elaborate on why data for 2020 were excluded for the sensitivity analysis.
- Pg3 Ln106: I am assuming you mean the number of foreign tourists increased? Please clarify.
- Pg3 Ln117: Please give actual number cases to demonstrate decrease in cases in 2019 compared the peak in 2000.
- Please remove the words “for the difference was” when giving the p value.
- Pg4 Ln127-128: Do you mean that as the numbers of Japanese nationals returning to Japan increased, so did the malaria risk? Please clarify.
- It would be nice to see the countries that the Japanese nationals who contracted malaria travelled to.
- Pg5 Ln145: Suggest a stronger opening sentence for the discussion that does not start with a verb.
- Pg5 Ln150: These finding suggested the reduction in leisure travel by Japanese nationals resulted in a decrease in malaria cases in Japan. Please state this clearly.
- Pg5 Ln157: Is there proof that Japanese doctors have problems in diagnosing malaria? Are there reported of delayed diagnosis?
- Pg5 Ln165: Please expand on why those who contracted malaria while overseas are more likely to progress to severe disease.
- Pg5 Ln183: Please provide a reference that China was declared malaria free in 2000
- Pg5 Ln197: The risk of malaria is equal among residents and visitors in high endemic areas. The only difference is that older residents have some level of immunity which makes that asymptomatic while non-immune visitors rapidly become symptomatic. Please ensure this point is clearly articulated.
- Any other recommendation other than improving malaria diagnosis in Japan. What about health education for traveler to malaria endemic regions or recommendations on chemoprophylaxsis?
Author Response
While this article reveals some interesting finding around the importation of malaria into Japan, the article requires significant revision before it can be considered for publication. I have listed a few comments below to assist with the revision. I also recommend that the article be reviewed by a English scientific editor before re-submission to ensure rules of scientific writing and English grammar have been adhered to.
>>Thank you very much for valuable comments. I revised my manuscript point by point as follows.
Pg1 Ln1-2: The article title needs to be reworded to more clearly articulate the aim of the article. I do not understand the term “international travel stagnation” and think most readers will not understand the phrase either. A time frame should be added to the title as well.
>> I revised the title as follows; “A model to estimate the effect of international traffic on malaria cases: a case of Japan from 1999 to 2021”
Define what you mean by “foreigners”. Are they tourists, students or non-Japanese nationals working in Japan?
>> I added the definition of “foreigners” in method section as follows; “Japanese was defined as “a person with Japanese nationality”, and foreigners was de-fined as “a person without Japanese nationality who were not “a person under the agreement” in this article. “A person under the agreement” is defined as “the foreign military forces, military civilian and their families who entered into or departed from Japan by civilian ships or aircrafts under the agreements relating to the status of U.S. and U.N. military forces stationed in Japan” in Statistics on Legal Migrants.” in L122-L128.
The introduction needs to be revised to ensure the readers have a clear idea of the malaria situation in Japan, whether malaria importation is a problem, the routes of malaria importation, COVID travel restrictions in Japan, what the research described in this article aimed to achieve and these possible uses of research findings.
>> I added the sentences of the malaria situation, whether malaria importation is a problem in Japan and the routes of malaria importation “In Japan, malaria was eradicated, and almost all cases were reported as imported malaria [14]. Persons aged 20-39 years accounted for 62% of cases [14], and the report by National Institute of Infectious Diseases concluded that this age distribution of malaria cases was associated with countries persons who infected malaria travelled to and age distribution of oversea travelers [14]. This report also showed that 688 cases of 704 cases between 2006 and 2017 were imported malaria, and persons who travelled to African countries accounted for 65% of cases [14]. These data suggest that malaria is rare in Japan, and a previous study suggested that there are few medical doctors with experience in diagnosing malaria in Japan [16].” in L67-L75, the paragraph of COVID travel restrictions in Japan “In the COVID-19 pandemic, government of Japan has been implementing risk-based border control by quarantine measures (e.g. submission of a certificate of negative test result conducted within 72 hours prior to departing from the country/region where travelers stay, stay for specified days at facilities designated by the Chief of the Quarantine Station, and stay for the remaining period of 14 days after the entry into Japan at places such as their own residence), denial of permission to entry of foreign nationals from specified country/area, and suspension of visa validity. On January 31st, 2020, Hubei Province was designated as the first region for denial of permission to entry of foreign nationals, after declaration of a Public Health Emergency of Inter-national Concern (PHEIC) under the International Health Regulations [1], and it was subsequently expanded to 73 countries/areas on April 3rd, 2020, after “announcement of pandemic” by WHO Director-General. As of the end of 2021, new entry of foreign nationals from all countries/areas is suspended for the time being, as emergency precautionary measure from a preventive perspective against Omicron variant. These border control measures affect the number of entries crossing international border of Japan and imported malaria cases in the COVID-19 pandemic.” in L94-L109, and the sentence of what the research described in this article aimed to achieve and these possible uses of research findings “prediction in future trend of malaria cases may assist medical doctors to aware of malaria risks in countries without indigenous malaria including Japan.” in L79-L91.
Terms like international traffic need to be explained – does this refer to cargo, business or leisure travel?
>>I added the definition of traffic as follows; “International traffic which is defined as “the movement of persons, baggage, cargo, containers, conveyances, goods or postal parcels across an international border, including international trade” in the International Health Regulation [1]” in L28-L30.
Pg1 Ln28: I do not understand what “the genealogy of these frameworks” means. Please rephrase more clearly.
>>I revised the sentence as follows; “In line with the historical frameworks,” in L33.
Pg1 Ln35: We are still very much in the midst of a COVID-19 pandemic – so please correct your statement about the being in the post-Corona era.
>>I revised the sentence as follows; “However, the situation changed completely after the beginning of pandemic of Coronavirus disease 2019 (COVID-19) [5,6].” in L39-L40.
Pg1 Ln36: I am assuming the word “all” is missing before the word “countries”
>>I added the word “all” as follows; “almost all countries have broadly applied interference with international traffic such as international travel restriction by border control [4,7].” in L41-L43.
Pg1 Ln39-40: Please stated whether these travel figures are for Japan, Asia or the global.
>>I added the words “in the world” as follows “An annual report by the World Tourism Organization (UNWTO) indicates that international tourist arrivals have fallen by 74% from 1.46 billion arrivals in 2019 to 381 million arrivals in 2020 in the world [8].” in L43-L45.
Pg1 Ln43-45: These relaxation in international travel was it leisure or business? What do you mean by minor uptick?
>>The data include leisure and business as arrivals. I added the description of the report by UNWTO as follows; “That is, UNWTO indicated that international tourism saw a minor uptick in May 2021 with arrivals declining by 82% (versus May 2019), after falling by 86% in April [10].” in L50-L51.
Pg2 Ln52: Please clarify what you mean by “ the identified area where cases infected have been estimated as oversea.”
>>I revised the sentence as follows; “almost all cases were reported as imported malaria [14]” in L58-L59.
Pg2 Ln58: What proof do you have that malaria cases will increase in Japan once covid-related travel restrictions are lifted?
>>If the number of entries is associated with the number of malaria cases, the rebound of the number of entries may increase the number of malaria cases. I added the sentence “as this change may be caused by an association between international traffic (i.e. the number of entries crossing international border) and imported infectious disease,” in L62-L64.
Pg2 Ln65: Where these entries for all international borders: land, air and sea?
>> In Japan, there is no international border of land, I added the sentence as follows; “report the number of all entries crossing international borders of Japan (i.e. airport and seaport).” in L121-L122.
Pg2 Ln81: Please add that as these publicaly available data were anonymized, informed consent was not required.
>> I revised the sentence as follows; “As these publicly available data were anonymized, informed consent was not required.” in L144-L145.
Pg3 Ln98: What guided the decision that VIFs above 5 indicted multicollinearity?
>>I added the sentences as follows; “This point of cutoff for VIFs is usually used for diagnosis of multicollinearity, and VIFs above 5 means that the R2 is 0.8 and more. When the R2 is 1, it means perfect collinearity [22-23].” in L162-L166.
Pg3 Ln100: Elaborate on why data for 2020 were excluded for the sensitivity analysis.
>> I added the sentence as follows; “Because both of entries and malaria cases were largely reduced after 2020, this large reduction after 2020 may affect the association of entries with malaria cases between 1999 and 2021.” in L165-L167.
Pg3 Ln106: I am assuming you mean the number of foreign tourists increased? Please clarify.
>> I added the sentence as follows; “Statistics on Legal Migrants shows that the increased number of entries from around 2012 was caused by increment of foreign tourists [9].” in L178-L190.
Pg3 Ln117: Please give actual number cases to demonstrate decrease in cases in 2019 compared the peak in 2000.
>> I added the sentence as follows; “The weekly number of malaria cases was 9 in the peak of 2000, but the respective number was 4 in the peak of 2019.” in L189-L190.
Please remove the words “for the difference was” when giving the p value.
>> I removed the words “”for the difference was” in L195
Pg4 Ln127-128: Do you mean that as the numbers of Japanese nationals returning to Japan increased, so did the malaria risk? Please clarify.
>> This sentence means that the high number of entries for Japanese (persons with Japanese nationals) was associated with the increased number of malaria cases. I revised as follows; “Table showed that the high number of entries for Japanese was associated with the increased number of malaria cases.” in L167-L169.
It would be nice to see the countries that the Japanese nationals who contracted malaria travelled to.
>> Because individual data of malaria cases such as countries persons who infected malaria travelled are not publicly available, I could not use these data. However, a previous study showed that persons who travelled to African countries accounted for 65% of cases [14]. I added the limitation as follows; ”A limitation of this article is that individual data of malaria cases such as age and countries persons who infected malaria travelled to were not used, because these data are not publicly available. A summary of malaria cases between 2006 and 2017 reported by National Institute of Infectious Diseases showed that people aged 20-39 years ac-counted for 62% of cases, and concluded that this age distribution of malaria cases was associated with countries persons who infected malaria travelled to and age distribution of oversea travelers [14]. This report also showed that 688 cases of 704 cases were imported malaria, and that persons who travelled to African countries accounted for 65% of cases [14].” in L322-L329.
Pg5 Ln145: Suggest a stronger opening sentence for the discussion that does not start with a verb.
>> I revised the sentence as follows; “The association of the number of entries for Japanese and foreigners with the number of malaria cases during past two decades in Japan was examined, aiming to identify the potentially reduced number of malaria cases by stagnation of international traffic after pandemic of COVID-19.” in L226-L229.
Pg5 Ln150: These finding suggested the reduction in leisure travel by Japanese nationals resulted in a decrease in malaria cases in Japan. Please state this clearly.
>> Because there are not statistics in purpose of international travel (e.g. leisure or business) for Japanese in Statistics on Legal Migrants, the association specified by purpose of international travel was not examined in this study. Therefore, I added the limitation as follows; “Second, as there is not statistics in purpose of international travel (e.g. leisure or business) for Japanese in Statistics on Legal Migrants, the association specified by purpose of international travel was not examined in this study. Because a previous study showed that malaria cases who traveled to countries with endemic malaria for tourism had higher mortality rate, compared to those for visiting friends and relatives [27], the purpose of international travel may also affect risk of infection. As individual data of entries as well as malaria cases may be helpful to make the predictive model more ac-curate, further studies using individual data are required for more accurate predictive models.” in L329-L337.
Pg5 Ln157: Is there proof that Japanese doctors have problems in diagnosing malaria? Are there reported of delayed diagnosis?
>> In a previous study, the delayed diagnosis was reported. I added the sentences as follows; “Moreover, a previous article suggested that there were “Patient’s delay” and “Doctor’s delay” in Japan, as there are few medical doctors with experience in diagnosing malaria in Japan [16].” in L280-L285.
Pg5 Ln165: Please expand on why those who contracted malaria while overseas are more likely to progress to severe disease.
>> I added the sentence as follows; “As naturally acquired immunity to malaria is usually developed by individuals living in a malaria-endemic area [17], visitors who infected malaria while overseas are more likely to be non-immune.” in L249-L251.
Pg5 Ln183: Please provide a reference that China was declared malaria free in 2000.
>> I cited the reference number 18.
Pg5 Ln197: The risk of malaria is equal among residents and visitors in high endemic areas. The only difference is that older residents have some level of immunity which makes that asymptomatic while non-immune visitors rapidly become symptomatic. Please ensure this point is clearly articulated.
>> I deleted this paragraph.
Any other recommendation other than improving malaria diagnosis in Japan. What about health education for traveler to malaria endemic regions or recommendations on chemoprophylaxsis?
>> I added the sentences regarding health education and chemoprophylaxis as follows; “We also need to raise awareness, and to promote health education of malaria risks as well as chemoprophylaxis for overseas travelers with normalization of international traffic.” in L258-L259.
Round 2
Reviewer 2 Report
Thanks for responding to my comments.
Reviewer 3 Report
No other comments, except to request that the word "stagnation" be replace with a more appropriate term such "sharp decline".